# Complexity of NAC Action as an Antidiabetic Agent: Opposing Effects of Oxidative and Reductive Stress on Insulin Secretion and Insulin Signaling

**DOI:** 10.3390/ijms23062965

**Published:** 2022-03-09

**Authors:** Lital Argaev-Frenkel, Tovit Rosenzweig

**Affiliations:** 1Department of Molecular Biology, Ariel University, Ariel 4070000, Israel; litalfren@gmail.com; 2Department of Nutritional Studies, Ariel University, Ariel 4070000, Israel

**Keywords:** type 2 diabetes, N-acetylcysteine, redox balance, glucose uptake, glutathionylation, oxidative stress, insulin signaling

## Abstract

Dysregulated redox balance is involved in the pathogenesis of type 2 diabetes. While the benefit of antioxidants in neutralizing oxidative stress is well characterized, the potential harm of antioxidant-induced reductive stress is unclear. The aim of this study was to investigate the dose-dependent effects of the antioxidant N-acetylcysteine (NAC) on various tissues involved in the regulation of blood glucose and the mechanisms underlying its functions. H_2_O_2_ was used as an oxidizing agent in order to compare the outcomes of oxidative and reductive stress on cellular function. Cellular death in pancreatic islets and diminished insulin secretion were facilitated by H_2_O_2_-induced oxidative stress but not by NAC. On the other hand, myotubes and adipocytes were negatively affected by NAC-induced reductive stress, as demonstrated by the impaired transmission of insulin signaling and glucose transport, as opposed to H_2_O_2_-stimulatory action. This was accompanied by redox balance alteration and thiol modifications of proteins. The NAC-induced deterioration of insulin signaling was also observed in healthy mice, while both insulin secretion and insulin signaling were improved in diabetic mice. This study establishes the tissue-specific effects of NAC and the importance of the delicate maintenance of redox balance, emphasizing the challenge of implementing antioxidant therapy in the clinic.

## 1. Introduction

Type 2 diabetes (T2D) is a chronic disease, recognized to be a risk factor for several severe morbidities, such as cardiovascular disease, stroke, neuropathy, nephropathy and retinopathy. The prevalence of T2D is high and is still on the rise, globally affecting large numbers of individuals. Despite the substantial increase in the number of antidiabetic drugs and the improvements in treatment protocols, a significant percentage of patients still fails to maintain acceptable blood glucose levels [1], emphasizing the need for new treatment strategies.

T2D is primarily characterized by insulin resistance, although as the disease progresses, the dysfunction and destruction of pancreatic beta cells also occur, leading to the deterioration of the ability to maintain glucose and lipids homeostasis. The development of T2D is driven by several factors, among them being oxidative stress, which is implicated as a contributing factor at all stages of the disease [2]. Oxidative stress is involved in the pathological processes leading to T2D by the activation of stress signals that attenuate the transmission of insulin signaling [3]. Oxidative stress also attacks pancreatic beta cells, which have a low capacity in the antioxidant machinery and thus are highly sensitive to such damage [4,5]. The resulting insulin resistance and beta-cell dysfunction lead to hyperglycemia which, along with accompanying metabolic inflammation, intensify the severity of oxidative stress. In turn, this stress worsens the pathology further with the additional deterioration of insulin sensitivity and its secretion. Finally, by attacking various biological components, oxidative stress plays a role in tissue damage, leading to diabetic complications [6]. Thus, oxidative stress is not only developed by the metabolic defects leading to T2D, but it also aggravates the defects and contributes to diabetic damages [7]. Accordingly, antioxidant (AOX) therapy is expected to negate oxidative stress to prevent the onset of T2D and to reduce the severity of the disease [8].

The potential efficacy of antioxidants is strongly supported by a large number of animal studies, demonstrating that antioxidants ameliorate oxidative stress and confer antidiabetic properties [9]. However, unfortunately, most clinical studies failed to provide adequate support for the use of antioxidants as a treatment or as an adjunct therapy in diabetes [10]. Among the antioxidants used in these studies, only lipoic acid was found to be effective [11], while vitamin E and C failed [12,13].

The failure of AOX therapy in human studies might be attributed to the two-faced reactive oxygen species (ROS) action, playing a role in both the physiology and pathophysiology of glucose homeostasis. Although damaging agents in high amounts, when applied at physiological levels, ROS play a role as second messengers and are required for a wide range of cellular functions, including the transmission of insulin signaling [14,15]. An example of the dose-dependent antagonistic nature of ROS action is their effects on cysteine thiols. The oxidation of cysteine thiolate residues to sulfenic acid is the main mechanism in which ROS participate in signal transduction. Such oxidation causes allosteric changes that affect protein–protein interaction and enzymatic function, leading to changes in the transmission of several signaling cascades [16]. At physiological ROS levels, thiolate–sulfenic oxidation is reversible by the action of disulfide reductases. However, at higher oxidative conditions, sulfenic acid is irreversibly oxidized further to sulfinic and sulfonic acids, resulting in permanent protein modification associated with oxidative stress [17].

The necessity to maintain a delicate redox balance complicates the ability to use AOXs as a treatment strategy in diabetes and other chronic conditions [18]. In addition, tissue-specific differences in redox balance regulation also contribute to the complexity of AOX therapy. The activity level of the AOX defense system and the rate of ROS generation are not equal in all cells [19,20,21]. Accordingly, tissues might differ in their vulnerability to oxidative stress, as well as their response to AOX supplementation. Thus, insulin-secreting pancreatic beta cells, which are highly vulnerable to oxidative stress, are impaired by moderate levels of ROS [22]. However, insulin signaling in adipocyte and myotubes might be less affected by the same dose of ROS. Similarly, the response to AOX and the effect of such treatment on redox balance and cell functionality are expected to be tissue-specific.

Our previous study demonstrated major improvements in glucose tolerance and insulin sensitivity among KK-Ay diabetic mice and high-fat-diet-fed C57bl/6J mice receiving dietary supplementation of NAC [23]. However, an inverted U-shape curve was observed in regard to NAC’s effect on insulin sensitivity, demonstrating its benefit at certain doses, while higher doses were ineffective. The aim of this study was to characterize the specific dose-dependent effects of NAC on different tissues which play a role in the regulation of blood glucose. The effects of NAC on insulin secretion were measured in isolated pancreatic islets, and the transmission of insulin signaling was measured in myotube and adipocyte cell lines. In addition, the different effects of NAC on insulin sensitivity in control and diabetic mice were demonstrated. This study established the tissue-specific effects of NAC and the development of reductive stress in response to NAC in certain cells and emphasized the challenge of implementing AOX therapy in the clinic.

## 2. Results

Glucose homeostasis is maintained by the concerted action of insulin-secreting cells and insulin-responsive tissues. In the first part of the study, the effects of H_2_O_2_ and NAC on pancreatic β cells were investigated. Pancreatic islets were isolated from untreated ICR mice and were treated ex vivo with increasing concentrations of H_2_O_2_ or NAC. NAC treatment did not affect the viability of the islets (Figure 1A) and their capability to secrete insulin (Figure 1B). On the other hand, H_2_O_2_ induced islet damage, leading to extensive cell death at a concentration of 100 μM. In addition, H_2_O_2_ inhibited basal insulin secretion, while glucose-induced insulin secretion was preserved (Figure 1C). These results are in line with previous studies, demonstrating the high vulnerability of pancreatic β cells to oxidative stress [22].

In the next part of the study, the effects of redox imbalance on insulin responsive cells were measured. For this, L6 myotubes and 3T3-L1 adipocytes were treated with NAC and H_2_O_2_ at increasing concentrations. The effects of such oxidizing and reducing conditions on cell viability were measured (Figure 2). Both H_2_O_2_ and NAC exerted cytotoxic effects when given at increasing concentrations in L6 myotubes (Figure 2A,B) and 3T3-L1 adipocytes (Figure 2C,D).

The activation of the insulin signaling cascade was measured using a glucose transport assay (Figure 3 and Figure 4) and Western blot analysis. Insulin treatment was performed as a positive control. Glucose uptake was measured in differentiated 3T3-L1, treated with H_2_O_2_ or NAC for 40 min or 24 h. H_2_O_2_ stimulated glucose transport in both the acute and chronic treatment conditions (Figure 3A and Figure 4A). However, while insulin-induced glucose transport was elevated by acute H_2_O_2_ administration, even when given at low concentrations (Figure 3C), this observation was not found when H_2_O_2_ was given for a prolonged time, and a reduction in insulin-induced glucose uptake was obtained under high oxidizing levels (Figure 4C). On the other hand, although basal glucose transport was not affected by an acute NAC treatment (Figure 3B), it was significantly inhibited by chronic exposure to the antioxidant (Figure 4B). In addition, both acute and chronic NAC treatments abolished insulin-induced glucose transport (Figure 3D and Figure 4D).

According to glucose uptake data, an inhibitory effect of NAC was demonstrated only when given for 18–20 h. In the next experiments, the effect of chronic NAC treatment was further investigated. The effects of chronic exposure to H_2_O_2_ and NAC on the activation of insulin signaling cascade in L6 and 3T3-L1 cells are demonstrated in Figure 5. As shown, NAC inhibited the insulin-induced phosphorylation of IR, Akt and ERK in 3T3-L1 adipocytes (Figure 5A). In addition to these proteins, GSK3β was also inhibited by NAC in L6 myotubes (Figure 5B).

Chronic treatment with H_2_O_2_ almost did not affect insulin signaling cascade in 3T3-L1 adipocytes, while stimulating GSK3β and ERK phosphorylation in L6 myotubes. The phosphorylation of JNK and p38, which are stress proteins of the MAPK family, was increased by H_2_O_2_ in both of the two cell lines, while NAC did not induce this kind of stress response, as determined by the lack of p38 and JNK phosphorylation.

In order to characterize whether the regulation of insulin signaling by these agents is accompanied by an alteration in redox balance, we measured the levels of GSH and GSSG, which are the reduced and oxidized forms of glutathione. Glutathione levels were measured in 3T3-L1 adipocytes (Figure 6A–C) and L6 myotubes (Figure 6D–F). As expected, NAC, a precursor for glutathione synthesis, increased all forms of glutathione, both oxidized and reduced (Figure 6). This effect was more evident in L6 myotubes, which were found to be more sensitive to NAC treatment than 3T3-L1. On the other hand, H_2_O_2_ did not affect glutathione levels in 3T3-L1 adipocytes but had a biphasic effect in L6 myotubes; while low-dose H_2_O_2_ increased GSH, higher concentrations of this oxidizing agent depleted glutathione pools, and both GSH and GSSG levels were reduced.

It was suggested that alterations in redox balance might affect cell function by inducing protein modifications. Thiol residues are highly sensitive to redox shift and might form disulfide bonds or undergo glutathionylation upon exposure to an oxidizing environment [24]. Accordingly, the level of free thiols, a biomarker of redox modifications of proteins, was measured. No difference in free thiol levels was observed in H_2_O_2_-treated cells, while an increase was induced by NAC in both 3T3-L1 adipocytes (Figure 7A) and L6 myotubes (Figure 7B).

An oxidative environment promotes the conjugation of glutathione to cysteine thiol groups (glutathionylation). The effects of NAC and H_2_O_2_ on protein glutathionylation was measured in L6 myotubes (Figure 8). The basal glutathionylation level of a ~45kDa protein was high in L6 myotubes and was not elevated further by H_2_O_2_, while it was reduced by NAC (Figure 8A). Insulin had no clear effect on such a modification. The identification of glutathionylated proteins was performed in control and NAC-treated L6 myotubes by mass spectrometry. NAC reduced the glutathionylation level of 206 proteins, while it increased the glutathionylation of only 63 proteins (Figure 8B). Biological functions of the proteins were characterized (Uniprot) and presented in the graph. Among affected proteins, cytoskeletal proteins and those involved in the machinery of translation (mainly ribosomal proteins) were prominent substrates for glutathionylation (either induced or repressed by NAC), noticeably, the NAC-induced de-glutathionylation of proteins of certain biological functions. These included metabolic enzymes, signaling proteins, proteins involved in protein trafficking and proteins involved in folding, repair and unfolded-protein response. Metabolic enzymes include enzymes of glycolysis, TCA cycle, ATP synthesis and glucosamine biosynthesis (Appendix A). Signaling proteins include a variety of proteins, of different groups, but are composed of a relatively large number of GTPases and isoforms of annexins and of the 14-3-3 family of proteins (Appendix A). Among proteins involved in trafficking (Appendix A), some are also small GTPases or GTP-binding proteins (ADP-ribosylation factor 1 and 4). The group of proteins involved in folding, repair and UPR contains a large number of heat-shock proteins, chaperones and isomerases regulating protein disulfides (Appendix A).

The results so far demonstrate that the viability and functionality of pancreatic islets are highly sensitive to the damaging effects of an oxidizing environment. On the other hand, the transmission of insulin signaling in L6 and 3T3-L1 cells was increased by H_2_O_2_ while being negatively affected by NAC. Thus, in view of the complexity of NAC outcomes on different tissues, in the last part of the study, the effects of NAC were investigated in vivo in both healthy and diabetic mice. In healthy, normoglycemic C57bl6 mice, NAC supplementation did not affect body weight accumulation, fasting blood glucose or glucose tolerance, although the phosphorylation of Akt, GSK3β and PRAS40 was attenuated in the liver of NAC-treated mice (Figure 9).

On the other hand, a beneficial effect was observed when given to KK-Ay mice, a model of type 2 diabetes. NAC (1200 mg/kg/day) improved glucose tolerance and increased the basal phosphorylation of signaling proteins of the insulin receptor cascade in liver and muscle (Figure 10), while the insulin-induced phosphorylation of these proteins was not affected (data not shown). In addition, the in vivo administration of NAC to diabetic KK-Ay mice stimulated the capability of isolated islets to secrete insulin under basal glucose conditions (Figure 10).

## 3. Discussion

The intimate link between oxidative stress and T2D is well established by a large number of studies [2]. Therefore, the failure of clinical trials to demonstrate the benefits of AOXs in the treatment of T2D was highly disappointing. Inappropriate research design and inadequate research population as well as incorrect outcome variables, specific antioxidant, or dose selection were blamed to cause the in-effectivity of AOX therapy [9,18,25,26]. In this study, we suggest that the tissue-specific requirements for AOX are another obstacle in the successful implementation of these agents into the clinic. Redox balance is differently maintained among tissues, reflecting the tissue-specific rate of ROS production and the activity of the AOX systems. Accordingly, the capability to maintain a delicate redox balance is different between distinct tissues. While one tissue might be highly vulnerable to an oxidizing agent, another tissue might be less affected. Thus, the administration of AOX might differently affect the redox balance in distinct tissues.

The dose-dependent effects of NAC and H_2_O_2_ on several cell functions were demonstrated. The concentration scale includes high doses of these agents (in the milli-molar range), which were also used in previous studies [25,26,27,28,29]. These doses of NAC and H_2_O_2_ induced a reductive or oxidative stress, respectively, in distinct tissues. Thus, while pancreatic islets are highly sensitive to oxidative stress, insulin signaling in responsive tissues is impaired by reductive stress. It is highly difficult to implicate in vitro concentrations in in vivo situations because of the complicated pharmacokinetic of the agents and the redox regulatory mechanisms which might neutralize oxidants and antioxidants. However, although the exact concentration might differ between in vitro and in vivo systems, the phenomenon demonstrated here of the tissue-specific effects of oxidative and reductive stress should be implicated in both.

A common misconception is that oxidants are mainly harmful agents that should be completely eliminated, while AOXs exert absolute beneficial outcomes on individual health, meaning “the more, the better”. In fact, ROS display hormesis and are essential intermediates playing a role as second messengers in cell signaling and in the regulation of several aspects of cell function [30,31,32]. Specific pathways are regulated by ROS in a tissue-specific manner; thus, the elimination of these agents might impair the functionality of certain tissues [17,33]. Specifically, the activation of insulin signaling elicits the generation of H_2_O_2_, which plays a role as a second messenger for the transmission of the cascade [27,33,34,35]. We confirmed that acute exposure to H_2_O_2_ stimulated basal and insulin-induced glucose transport. On the other hand, chronic exposure to H_2_O_2_ inhibited insulin-induced glucose uptake. The mechanism that may be involved in these findings might include redox modifications of key signaling proteins leading to the activation of the insulin signaling cascade [31,36] even with the absence of insulin.

Several signaling proteins of the insulin receptor cascade are regulated by oxidation and might explain our observations. The insulin receptor (IR) itself is susceptible to stimulatory oxidation to the generation of disulfide (Cys1245–Cys1308) in its kinase domain [37]. In addition, IR phosphorylation is attenuated by the activity of protein tyrosine phosphatase (PTP1B), which is a well-known target for redox regulation [34]. A highly interesting aspect is the redox-dependent fine-tuning of AKT activity. AKT is susceptible to oxidation through the generation of disulfide located in its PH domain (Cys60–Cys77), which is required for the recruitment of the kinase to the membrane, where it is phosphorylated and activated [35]. This is in line with the stimulatory effect of a mild oxidative state on insulin signaling. However, AKT is inhibited at higher oxidizing conditions as a result of disulfide generation between Cys297–Cys311 within the kinase domain [38]. Hence, a delicate redox balance might be maintained for the optimal activation of AKT.

On the other hand, chronic exposure to relatively high levels of ROS has been associated with impaired physiological function through the cellular damage of macromolecules, leading to certain human pathologies including T2D [30,31]. As expected, our results demonstrate that H_2_O_2_ activates JNK and p38, which are key players in stress response. In addition, following chronic exposure (24 h), the stimulatory effect of H_2_O_2_ on insulin signaling was abrogated, as Akt phosphorylation was completely abolished, while the activation of the MAPK signaling pathway remained high in both 3T3-L1 adipocytes and L6-myotubes. The induction of glucose transport by H_2_O_2_, obtained under such a stress condition, might be mediated by Glut1 or other non-insulin-dependent glucose transporters, rather than Glut4-dependent transport [39,40]. Hence, our results support previous observations [41] of the bi-directional effect of oxidants on insulin-responsive cells, shifting from oxidative signaling to oxidative stress, in a dose-dependent manner.

Yet, our study demonstrated that the dose-dependency of oxidants is tissue specific. Cytotoxic effects were observed in adipocytes and myotubes treated with 1 mM H_2_O_2,_ which is a high dose of the oxidant. However, cell death was induced in pancreatic islets using much lower concentrations of this oxidizing agent. In addition, low H_2_O_2_ concentrations, which were found to stimulate glucose uptake in adipocytes, led to the dysfunction of pancreatic β cells. Furthermore, redox balance was differently maintained in various cell types; L6-myotubes showed the depletion of glutathione upon chronic exposure to H_2_O_2_, while 3T3-L1 showed an increase in glutathione levels under moderate oxidizing conditions, compared to untreated cells. Thus, the same dose of oxidizing agent differently affects redox potential and cell function in distinct tissues. These results illustrate the cell-specific capability to cope with alterations in redox balance; while some cells are relatively resilient to oxidation, other cells demonstrate high vulnerability to oxidative damage.

The theory of oxidants as required physiological agents was further supported by our data on NAC effects. For the first time, here, we demonstrated that increasing levels of NAC, a reducing agent, had negative outcomes on insulin signaling and glucose transport in L6 and 3T3-L1 and on the viability of these cells. The induction of cytotoxicity by NAC is in accordance with previous evidence of reductive-stress-induced cell death [42,43]. These data demonstrate that exaggerated AOX administration might be harmful to L6 myotubes and 3T3-L1 adipocytes, just as oxidants are damaging agents to pancreatic islets. Our data demonstrate that while some cellular systems might be damaged by oxidative stress, reductive stress might impair others, and these data emphasize the importance of identifying the intermediate dose, which prevents oxidative stress while not shifting the balance toward reductive stress.

Our in vivo data also support the sensitivity of insulin signaling to redox imbalances, both oxidative and reductive stresses. In accordance with previous observations [23,44,45], NAC improved glucose tolerance in diabetic mice. This improvement was accompanied by the enhancement of basal insulin signaling. However, although not affecting blood glucose levels, insulin signaling was impaired by NAC in control, normoglycemic mice. Thus, not only oxidative stress, but also reductive stress might impair insulin action.

Regarding the effects of redox imbalances on pancreatic islets, these were found to be highly sensitive to oxidative stress, displayed by cellular death and the repression of insulin secretion in response to even low doses of H_2_O_2_. On the other hand, while being highly damaged by oxidants, increasing the level of NAC, a reducing agent, did not impair these measures of islet function. These results suggest that NAC administration might be beneficial for the prevention of beta-cell deterioration in T2D patients. In addition, they are in line with most studies in the field demonstrating the negative effects of oxidants on beta cell function and the benefits of combatting oxidative stress in these cells [4,5,46,47]. In this study, we demonstrated that treatment with a reducing agent, even in the absence of oxidizing conditions, did not affect beta cell function. While not inducing damage, the stimulation of insulin secretion was also not observed in response to NAC, differing from a previous study demonstrating the facilitation of insulin secretion by compounds that include cysteine moiety [48]. The difference in results might be attributed to the different research model used. While most studies emphasize the sensitivity of beta cells to oxidative stress, there is some evidence that ROS at certain levels are required for proper function of insulin secretory machinery, suggesting that in pancreatic islets, redox regulation is also required, not only to prevent oxidative stress but also to avoid the depletion of ROS [49,50]. In this study, no negative outcome was observed in response to all NAC doses used; thus, this hypothesis was not supported by our research.

What is the mechanism mediating the inhibitory effect of NAC on insulin signaling?

NAC is an acetylated form of L-Cysteine and is a well-established AOX [51]. It is widely accepted that the major antioxidative attribute of NAC is supplying thiol (sulfhydryl) moieties for incorporation into protein chains and metabolites, particularly for the formation of glutathione, the most important intracellular antioxidant [44]. In this regard, NAC serves as an indirect antioxidant by replenishing the GSH pool, which is often depleted in oxidative stress associated with several pathologies [45,46,47,52,53]. In addition, NAC is a disulfide-breaking agent [54] with higher reducing efficiency compared with Cys and GSH [55,56]. This direct effect of NAC is responsible for its well-known mucolytic activity [57]. The thiolate of cysteine residues plays a role as redox switches, regulating protein activity. Under oxidizing conditions, thiol groups of redox-sensitive cysteines undergo oxidation in the generation of disulfide bonds with other protein thiols [58]. On the other hand, reducing conditions impairs protein disulfide formation. Data from in vitro studies suggest that NAC may alter protein structure and/or function by reducing disulfide bonds, thereby influencing the redox environment of the cell [59].

In addition to its effect on protein disulfides, NAC might affect protein folding, function and localization by modifying protein S-glutathionylation. This kind of a reversible post-translational modification might occur under oxidative stress, being a defensive event that protects thiolate from irreversibly damaging oxidation [60]. However, because protein s-glutathionylation is observed not only in stressed cells, it is postulated that this modification is a redox switch, playing a role in redox signaling [61]. There is still very little data regarding the regulation of glutathionylation and its effects on certain proteins, although it was reported that while some proteins were inhibited by S-glutathionylation, others were activated [62].

We found that NAC administration induced an elevation in GSH, an increase in free thiols and an alteration, mainly a reduction, in s-glutathionylation. While these functions of NAC are generally considered beneficial, there is evidence demonstrating some negative outcomes of these antioxidative properties. An increase in GSH level was observed in the adipose tissue of diabetic ob/ob mice, and the administration of GSH inhibited the insulin-induced phosphorylation of IRS-1 and Akt in 3T3-L1 adipocytes. Thus, GSH might be involved in insulin resistance in these cells [63]. The mechanism mediating this negative outcome of GSH accumulation has yet to be clarified, but a reduction in disulfide bonds in proteins might be involved [55]. These data are in line with our findings, demonstrating the adverse effects of NAC, a precursor of glutathione, on cell viability and insulin signaling, which were accompanied by an increase in GSH and free thiol levels.

The effects of NAC on the glutathionylation landscape might also be involved in its complexed effect on cell function. We found that NAC mostly reduced the level of glutathionylation and that the most represented classes of proteins among S-glutathionylated proteins were glycolytic enzymes and other energy-metabolizing enzymes, heat shock proteins, chaperones and protein disulfide isomerases, signaling proteins, translation factors and cytoskeletal proteins. This is in accordance with data obtained from large-scale proteomic studies [62]. However, the glutathionylation profile was mostly characterized under oxidative conditions [64,65], while the effect of reductive stress has yet to be investigated, to the best of our knowledge. Interestingly, it was reported that reductive stress impairs the activation of unfolded protein response, ER stress [66,67] and energy metabolism [68,69]. The high representation of proteins involved in such functions among those found to undergo deglutathionylation by NAC suggests that this modification plays a role in the adverse outcome of reductive stress.

The consequence of protein s-glutathionylation on protein function is complicated, and much data are still missing, although the fact that NAC affects the glutathionylation of so many proteins of diverse classes emphasizes the complexity of NAC action. Hypothetically, the impaired function of signaling proteins, i.e., the 14-3-3 proteins, which were de-glutathionylated by NAC, might lead to the suppression of insulin signaling [70]. In addition, disturbance in cytoskeletal assembly or in the machinery of intracellular transport might attenuate glucose transport in response to NAC. Small GTPases were found to be de-glutathionylated by NAC. The activity of this class of proteins was found to be either up- or down-regulated by glutathionylation [71,72]. These proteins are involved in actin remodeling and in Glut4 translocation [73] and thus might be involved in the mechanism of the NAC-induced inhibition of glucose transport. Similarly, ARF6, found to be de-glutathionylated by NAC, is involved in GLUT4 translocation [74]. Future studies should be performed in order to characterize the functional implication of the s-glutathionylation of specific proteins.

In view of this study, one may wonder if there is a future for NAC in the clinic. A number of studies reported the potential effects of NAC supplementation against the development of insulin resistance, T2D and its complications [75,76,77]. A protective effect of NAC was observed in db/db mice, among which, NAC treatment was associated with preserved insulin content, increased insulin mRNA as well as higher nuclear levels of pancreatic Pdx-1 [78]. The properties of NAC as an AOX that efficiently neutralizes ROS are well characterized. It was already shown that NAC counteracts the damage of ROS generated in response to H_2_O_2_ [79], dexamethasone [80], D-Glyceraldehyde [27], oleate [81] or chronic glucotoxicity [82]. In addition, NAC ameliorated HFD-induced elevation in mitochondrial and intracellular ROS, DNA and protein oxidative damage and adipose tissue inflammation and improved glucose and insulin tolerance [75]. Our previous study also supports the benefit of NAC administration for the management of glucose tolerance, although the importance of the eliminating overconsumption of NAC, which reduces its efficacy, was demonstrated [23]. Here, we show that while NAC facilitated insulin signaling in diabetic mice, a decrease in insulin signaling was observed in healthy mice. This difference might be attributed to the varying redox balance between diabetic and healthy conditions. NAC is beneficial in conditions of oxidative stress, such as T2D, while it might lead to a reductive redox imbalance in healthy mice.

NAC has the potential to preserve insulin secretion by preventing oxidative damage in pancreatic beta cells, although we assume that, when considering its benefits on insulin sensitivity, the effect of NAC is highly dependent on its dosage, which should be adapted to the current oxidative balance of the patient, as the overconsumption of NAC might impair insulin transmission in target tissues.

In summary, by presenting the pros and cons of NAC action, this research explains at least part of the failure of clinical studies with AOXs. Understanding the complexity of NAC action is the first and most essential step in order to develop strategies to overcome this obstacle for learning the benefits of NAC. Although a double-edged sword of sorts, this NAC action might be implicated in some other AOXs [82] and should be further validated in future studies.

We conclude that NAC might either improve or impair glucose balance, depending on the specific dose, which, rather than being a fixed one, might depend on the specific redox status of the tissue, and the overall redox balance of the individual. With that understanding, it is clear that the lack of a measurable biomarker for the presence and severity of oxidative stress is a major obstacle in the implementation of NAC into clinical use. Today, there are no commonly accepted biomarkers and guidelines to optimize a protocol of treatment with NAC or other AOXs for the treatment of T2D. Moreover, most clinical trials with AOXs lack the assessment of baseline oxidative stress in participants [83]. Accordingly, future research should focus on the characterization of biomarkers for redox balance, to enable better personalized adjustment of the AOX treatment. Such biomarkers should be measured to identify the individuals that might gain advantage from AOX supplementation and should be monitored at certain time-points during treatment in order to prevent reductive stress, which was shown in our research to ameliorate insulin signaling.

## 4. Materials and Methods

### 4.1. Materials

NAC was purchased from Calbiochem (Merck, Darmstadt, Germany). H_2_O_2_, insulin, proteases and phosphatase inhibitors, Collagenase-P and Histopaque, were all purchased from Sigma (Merck, Darmstadt, Germany). Anti-phospho Akt (S473), anti-Akt, anti-phospho-GSK (S9), anti-phospho IR (Y1150/1151), anti-IR and anti-phospho-PRAS40 (T246) were all purchased from Cell-signaling Technology (Danvers, MA, USA). Anti-β-tubulin was purchased from Abcam, and secondary antibodies (Peroxidase-AffiniPure Goat Anti-Mouse IgG antibody and Peroxidase-AffiniPure Goat Anti-Rabbit IgG antibody) were purchased from Jackson ImmunoResearch Laboratories (Baltimore Pike, PA, USA).

### 4.2. Methods

#### 4.2.1. In Vivo Studies

Mice were housed in an animal laboratory with a controlled environment of 20–24 °C, 45–65% humidity and a 12 h light/dark cycle. Animal House operates in compliance with the rules and guidelines of the Israel Council for Research in Animals, based on the US NIH Guide for the Care and Use of Laboratory Animals. The protocol was approved by the Committee on the Ethics of Animal Experiments of the University of Ariel (Permit Number: IL 76-09-15).

The interventional study was performed on KK-Ay mice (Jackson Laboratories, Bar Harbor, Maine, USA), a genetic model of T2D and on C57bl/6 mice (Envigo, Jerusalem, Israel). For these experiments, 6-week-old male mice were separated into treatment groups, 8 ± 10 mice each. KK-Ay and C57Bl/6J male mice were separated into 2 groups for each research model as follows: control, untreated mice and NAC-treated mice. The mice were fed a standard (STD) diet (18% of total calories derived from fat, 24% from proteins and 58% from carbohydrates. Harlan, Teklad TD.2018). The average consumption of water was measured and found to be 10 mL/day in both control and NAC-treated mice. NAC was dissolved in the drinking water so that the mice would receive NAC at about the dose of 1200 mg/kg of their body weight daily (3.6 mg/mL for mice of 30 gr), starting at age 6 weeks. Body weight was measured once a week. At the age of 12 or 15 weeks in the C57Bl/6J or KK-Ay, respectively, a glucose tolerance test (GTT) was performed. A week later, mice were anesthetized using ketamine + xylazine and euthanized via terminal bleeding followed by cervical dislocation. In order to follow insulin-induced protein phosphorylation in liver and skeletal muscle, in some of the mice (*n* = 5), insulin was injected (0.75 mU /g body weight) 15 min before killing the animal. Liver and muscle were snap frozen in liquid nitrogen and preserved at −80 °C for later protein extraction.

#### 4.2.2. Glucose Tolerance Test (GTT)

Mice were injected with 1.5 mg glucose/g body weight after 6 h fast. Blood glucose was determined from tail blood using the ACCU-CHEK Go glucometer (Roche, Munich, Germany).

#### 4.2.3. Islets Isolation

A protocol of islet isolation was conducted on control and NAC-treated KK-Ay (15 weeks old) and on ICR mice (10–14 weeks old). ICR mice (Swiss albino outbred strain) were purchased from Envigo Laboratories (Israel). Mice were anesthetized using a ketamine/xylazin mixture and killed via cervical dislocation. Pancreatic islets were isolated via the collagenase digestion of exocrine tissue, as described in [84]. Briefly, collagenase-P (1.4 mg/mL) was injected through the bile duct to the exocrine pancreatic ducts to the inflation of the pancreas. The pancreas was isolated and incubated for 15 min in 3 mL of collagenase solution at 37 °C. Cold RPMI was added to the mixture, and samples were centrifuged at 4 °C at 200× *g* for 1 min; then, supernatant was discarded. This washing step was repeated twice more. Islets were separated by gradient using Histopaque solutions (Merck) and 20 min of centrifugation at 4 °C at 1250× *g*. Separated islets were collected under a dissecting microscope into warm RPMI culture media and incubated in an incubator (37 °C, 5% CO_2_). Islet diameter was measured, and glucose-stimulated insulin secretion was performed following recovery of 24 h.

#### 4.2.4. Glucose Stimulated Insulin Secretion (GSIS)

The function of isolated islets was measured by analyzing GSIS. In this assay, isolated islets were incubated in a plate with a basal working solution of 3.3 mM glucose in KRB buffer at 37 °C for 30 min. Then, islets were transferred to a second plate with the same warm basal glucose solution for an additional 30 min. Islets were transferred to vials (10 islets in each vial) containing low- and high-glucose solutions (3.3 mM and 16.7mM, respectively), for one-hour incubation at 37 °C with moderate shaking. A gentle spin down of the islets was performed, supernatant was collected, and insulin was measured using an ELISA assay (Mercodia, Uppsala, Sweden), according to the manufacturer’s instructions. The results were measured using a microplate reader (Tecan, Männedorf, Switzerland) at a wavelength of 450 nm.

#### 4.2.5. Cell Culture

3T3-L1 pre-adipocytes (ATCC, passage number < 15) were cultured and induced to differentiate as described before [85]. L6 myoblasts (ATCC, passage number < 25) were grown in MEM-α containing 25 mM glucose, 10% FCS, 2 mM glutamine and 1% ampicillin. Experiments were performed on differentiated myotubes. L6 differentiation was induced as described in our previous studies [85].

#### 4.2.6. Glucose Uptake

To measure the short-term effects of NAC and H_2_O_2_, differentiated 3T3-L1 adipocytes were pre-incubated in serum-free DMEM for 2 h, followed by 30 min of treatment with H_2_O_2_ or NAC, with or without insulin (100nM) given for an additional 10 min. Cells were washed twice with phosphate-buffered saline (PBS) and incubated in PBS solution containing 0.1 mM 2-deoxy glucose (2DG) and 0.5 μCi [3H]-2DG for 5 min in 37 °C. Cytochalasin-B (20 µM) was used for the measurement of non-specific glucose uptake. Cells were then washed 3 times with cold PBS and lysed with 1% SDS solution. The contents of each well were transferred to a different plate containing Optiphase scintillation liquid (Perkin-Elmer) and counted using a MicroBeta counter (Perkin-Elmer). For long-term (chronic) effects, differentiated 3T3-L1 adipocytes were incubated with H_2_O_2_ or NAC for 18–20 h and then starved for 2 h in serum-free DMEM. Glucose uptake was then measured as described, with or without insulin treatment (100 nM) in the last 10 min.

#### 4.2.7. Glutathione Levels and GSH/GSSG Ratio

3T3-L1 adipocytes and L6 myotubes were treated for 18 h with H_2_O_2_ or NAC. Cells (10^7^–10^8^ cells) were washed with ice-cold PBS and harvested with lysis buffer of phosphate-buffered saline (PBS) containing 0.5% NP-40. After the homogenization, centrifugation and collection of the supernatant, deproteinization was performed by the addition of 1 volume TCA (100% *w*/*v*) into 5 volumes of sample. Samples were incubated for 5 min, and proteins were separated by centrifugation of 12,000 rpm for an additional 5 min. Acidity was neutralized using NaHCO_3_ to PH 4-6, and samples were centrifuged (13,000× *g* for 15 min at 4 °C). Supernatant was collected, and total glutathione and GSH were measured using a commercial kit (GSH/GSSG ratio detection fluorometric assay kit, Abcam, Cambridge, UK) according to the manufacturer’s protocol. The results were measured with a fluorescence microplate reader at Ex/Em of 490/520 nm compared to the standard curve. The GSSG level was calculated as follows: GSSG = (total glutathione-GSH)/2, enabling the calculation of the GSH/GSSG ratio as well.

#### 4.2.8. Protein Free Thiols (Ellman’s Assay)

3T3-L1 adipocytes and L6 myotubes were treated for 18 h with H_2_O_2_ or NAC. Cells were washed with ice-cold PBS, harvested with lysis buffer (PBS containing 0.5% NP-40), homogenized and centrifuged. The supernatant was collected, and the level of free thiols was measured using Ellman’s assay. This method is based on the ability of free thiols in a sample to reduce DTNB (Ellman’s reagent) by an exchange reaction, which forms mixed protein disulfides and TNB. TNB can be detected at 412 nm, and its levels are proportional to the reduced thiol content of the sample.

Samples (10 μL) were added to a reaction mix, containing 50 μL of DTNB solution (50 mM sodium acetate and 2 mM DTNB in water), 100 μL of Tris solution (1M Tris, pH 8.0) and 840 μL of water. The mix was incubated for 5 min at room temperature, and the optical density was measured using a microplate reader (Tecan). The molarity of -SH groups in the samples was quantitated by reference to the extinction coefficient of TNB (13,600 M^−1^ cm^−1^).

#### 4.2.9. Western Blot Analysis

Differentiated L6 and 3T3-L1 cells were treated with NAC or H_2_O_2_ for 24 h with or without insulin administration (100 nM) for the last 10 min. Protein lysates were prepared using RIPA buffer supplemented with protease and phosphatase inhibitors. The samples were homogenized and centrifuged at 14,000 rpm for 20 min. The supernatant was collected, and protein concentration was measured using the Bradford method. Protein was separated (20 μg per lane) via SDS-polyacrylamide gel electrophoresis. Proteins were electrophoretically transferred onto nitrocellulose membranes, which were blocked in 5% dry milk and incubated with the appropriate antibody solutions (5% BSA in 0.01% TBST). Proteins were immunodetected using the enhanced chemiluminescence method.

#### 4.2.10. Mass Spectrometry

L6-myotubes were treated with NAC (10 mM) for 24 h and harvested and lysed with lysis buffer (50Mm Tris pH 7.4 with 150Mm NaCl and Triton 1% added with protease inhibitor 1:100) with or without the addition of the reducing agent DTT (10 mM). Cell lysates were incubated with GSH-antibody (Abcam, 1:100) at 4 °C for 1 h. Pre-washed protein G/protein A-agarose bead mixture (Merck) was added to protein–antibody complexes and incubated at 4 °C for 30 min. Excess unbound proteins were washed with PBSx1 four times, and GSH–protein complexes were eluted (50 mM Tris, Ph 7.5, 5% SDS). Mass spectrometry profiling of eluted proteins was conducted at the Nancy and Stephen Grand Israel National Center for Personalized Medicine (Weizmann Institute of Science, Rehovot, Israel). The samples were digested with trypsin using the S-trap method. The resulting peptides were analyzed using nanoflow liquid chromatography (nanoAcquity) coupled to high-resolution, high-mass-accuracy mass spectrometry (Fusion Lumos). Each sample was analyzed on the instrument separately in a random order in discovery mode. Raw data were processed with MaxQuant v1.6.6.0. The data were searched with the Andromeda search engine against the Rat proteome database (SwissProt Nov19) appended with common lab protein contaminants and the following modifications: fixed carbamidomethylation on C, variable protein N-terminal acetylation, oxidation on M and deamidation on NQ. Quantification was based on the LFQ method, based on unique peptides. Data of reduced (DTT-treated) samples were subtracted for the non-reduced samples.

#### 4.2.11. Statistical Analysis

Values are presented as mean ± SEM. Statistical differences between the treatments and controls were tested by unpaired two-tailed Student’s *t*-test or one-way analysis of variance (ANOVA), followed by Bonferroni’s post hoc testing when appropriate. Analysis was performed using the GraphPad Prism 8.0 software. A difference of *p* = 0.05 or less in the mean values was considered statistically significant.

## Figures and Tables

**Figure 1 ijms-23-02965-f001:**
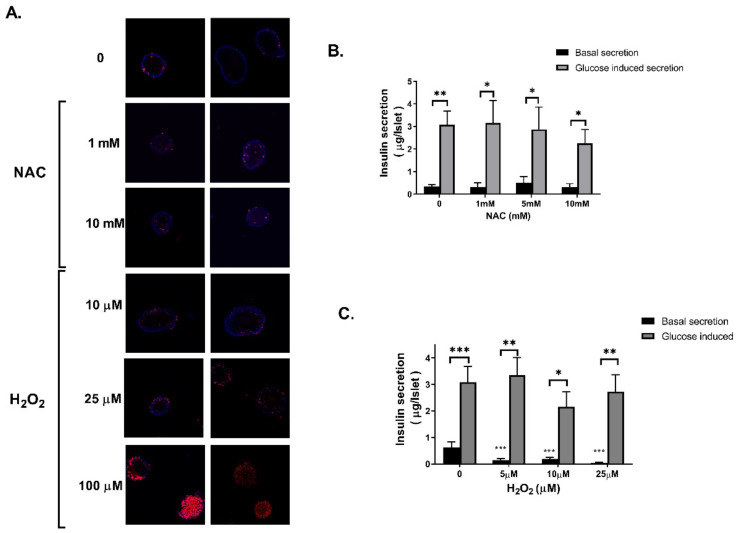
Effect of NAC on viability and functionality of pancreatic islets. Pancreatic islets were isolated from ICR mice and treated with NAC or H_2_O_2_ for 24 h. Cytotoxicity was detected using PI staining. Hoechst staining indicates viable cells. Two representative micrographs are presented for each treatment (**A**). (**B**,**C**): Basal and glucose-induced insulin secretion were measured as described in Methods. The data represent the mean ± SEM of 5 independent experiments. * *p* < 0.05, ** *p* < 0.005 and *** *p* < 0.0005, compared to control, or as indicated, analyzed by two-way ANOVA, followed by Bonferroni’s post hoc testing.

**Figure 2 ijms-23-02965-f002:**
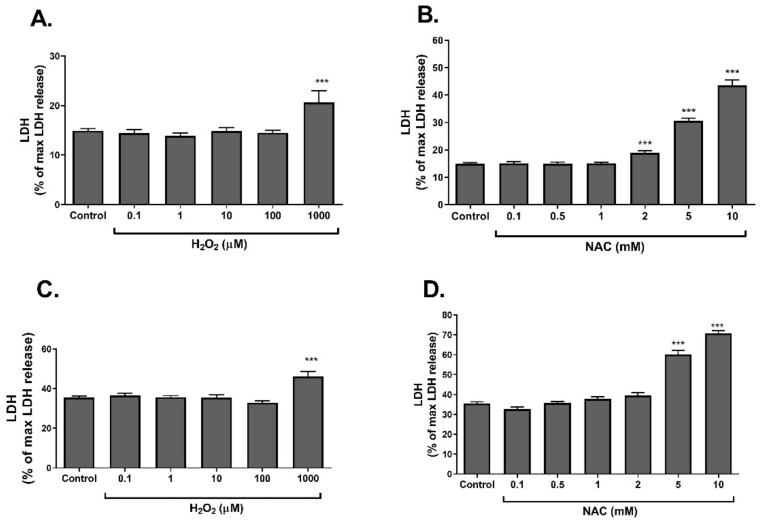
Cytotoxic effects of H_2_O_2_ and NAC. L6 myotubes (**A**,**B**) and 3T3-L1 adipocytes (**C**,**D**) were treated with H_2_O_2_ or NAC for 24 h. LDH release was measured as a biomarker of cytotoxicity, as described in Methods. The data represent the mean ± SEM of 4 independent experiments. *** *p* < 0.0005, compared to control, analyzed by one-way ANOVA, followed by Bonferroni’s post hoc testing.

**Figure 3 ijms-23-02965-f003:**
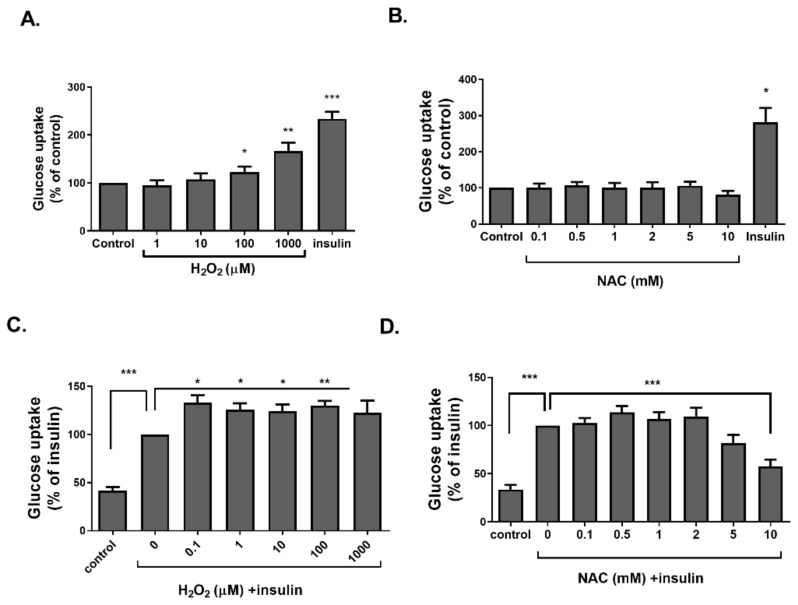
Short-term NAC treatment did not affect, while H_2_O_2_ increased glucose uptake in 3T3-L1 adipocytes. Differentiated 3T3-L1 adipocytes were treated with H_2_O_2_ (**A**) or NAC (**B**) for 30 min. Insulin (100 nM) was given as a positive control. In a second set of experiments, cells were treated with H_2_O_2_ (**C**) or NAC (**D**) for 10 min before the administration of insulin (100 nM) for an additional 30 min. Deoxy-D-glucose in cells was determined as described in Methods. Data are expressed as percent of basal uptake in control cells. The data represent the mean ± SEM of 5 independent experiments. * *p* < 0.05, ** *p* < 0.005 and *** *p* < 0.0005, compared to control or as indicated, analyzed by one-way ANOVA, followed by Bonferroni’s post hoc testing.

**Figure 4 ijms-23-02965-f004:**
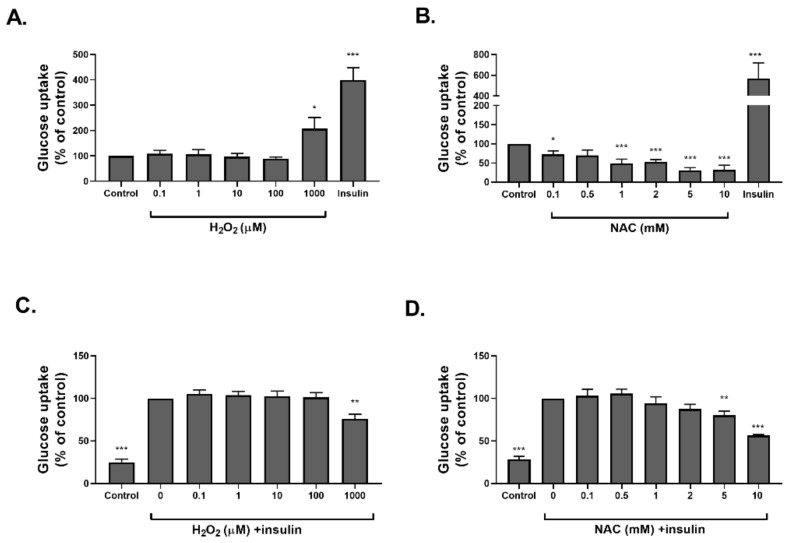
Long-term NAC treatment reduced glucose uptake in 3T3-L1 adipocytes. Differentiated 3T3-L1 adipocytes were treated with H_2_O_2_ (**A**) or NAC (**B**) for 24 h. In a second set of experiments, cells treated with H_2_O_2_ (**C**) or NAC (**D**) for 24 h were incubated with insulin (100 nM) for an additional 30 min. Deoxy-D-glucose in cells was determined as described in Methods. Data are expressed as percent of basal uptake in control cells. The data represent the mean ± SEM of 5 independent experiments. * *p* < 0.05, ** *p* < 0.005 and *** *p* < 0.0005, compared to control or as indicated, analyzed by one-way ANOVA, followed by Bonferroni’s post hoc testing.

**Figure 5 ijms-23-02965-f005:**
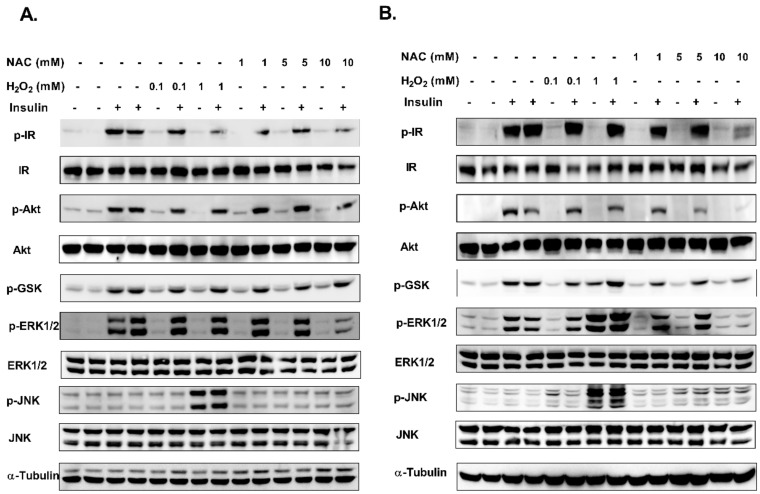
NAC reduced insulin-induced phosphorylation of signaling proteins in 3T3-L1 adipocytes and L6 myotubes. 3T3-L1 (**A**) and L6 (**B**) were treated with H_2_O_2_ or NAC for 24 h (**B**) followed by incubation with insulin (100 nM) for an additional 10 min. Western blot analysis of proteins involved in insulin signaling pathway was performed. Representative blot of 3 independent experiments is shown.

**Figure 6 ijms-23-02965-f006:**
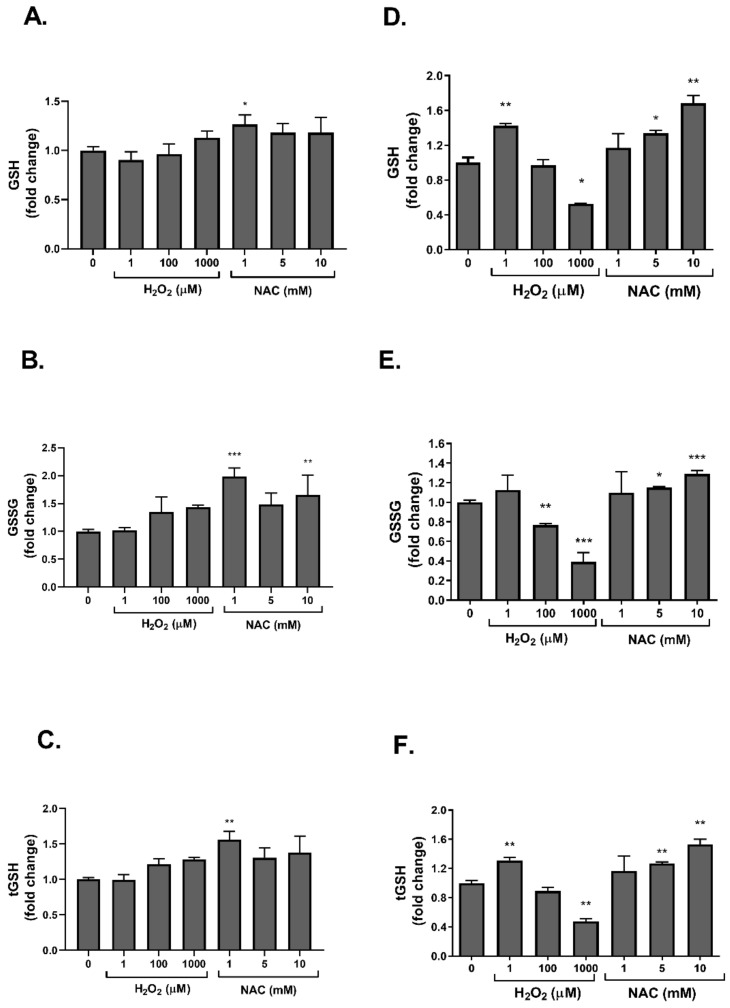
NAC increased glutathione level in 3T3-L1 adipocyte and L6 myotubes. 3T3-L1 (**A**–**C**) and L6 myotubes (**D**–**F**) were treated with H_2_O_2_ or NAC for 24 h. Levels of GSH, GSSG and total glutathione (tGSH) were measured as described in Methods. The data represent the mean ± SEM of 5 independent experiments. * *p* < 0.05, ** *p* < 0.005 and *** *p* < 0.0005, compared to control, analyzed by one-way ANOVA, followed by Dunnett’s post hoc testing.

**Figure 7 ijms-23-02965-f007:**
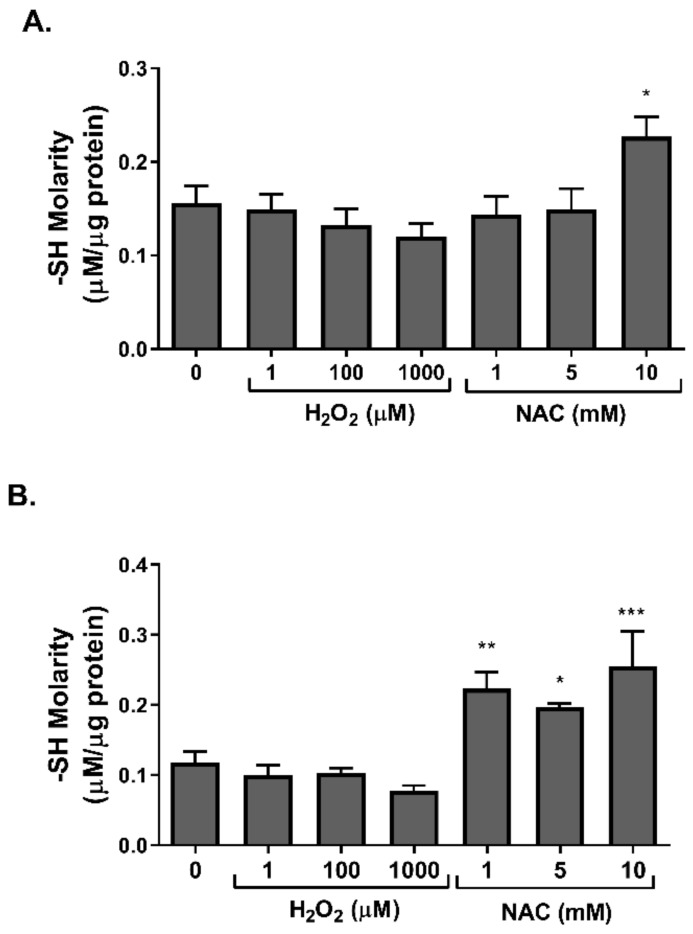
NAC increased the level of free thiol residues in 3T3-L1 and L6 cells. 3T3-L1 adipocytes (**A**) and L6 myotubes (**B**) were treated with H_2_O_2_ or NAC for 24 h. Free thiol (SH) level was measured according to Elmann’s assay, as described in Methods. The data represent the mean ± SEM of 5 independent experiments. * *p* < 0.05, ** *p* < 0.005 and *** *p* < 0.0005, compared to control, analyzed by one-way ANOVA, followed by Dunnett’s post hoc testing.

**Figure 8 ijms-23-02965-f008:**
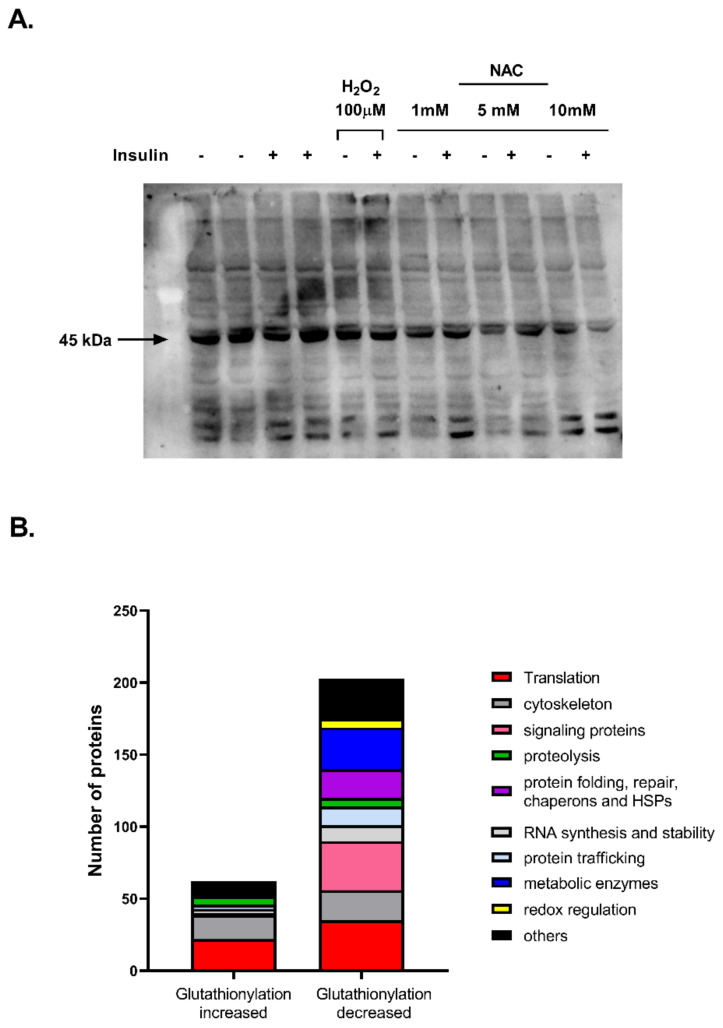
Protein glutathionylation is reduced by NAC in L6 cells. L6 myotubes were treated with H_2_O_2_ or NAC for 24 h. (**A**) Western blot analysis was performed under non-reducing conditions using anti-GSH antibody. Representative blot of 3 independent experiments is shown. (**B**) Number and classification of proteins with glutathionylation level that was increased or decreased by NAC. Level of glutathionylation was detected in L6 myotubes by mass spectrometry as described in Methods. Function of affected proteins was determined using Uniprot.

**Figure 9 ijms-23-02965-f009:**
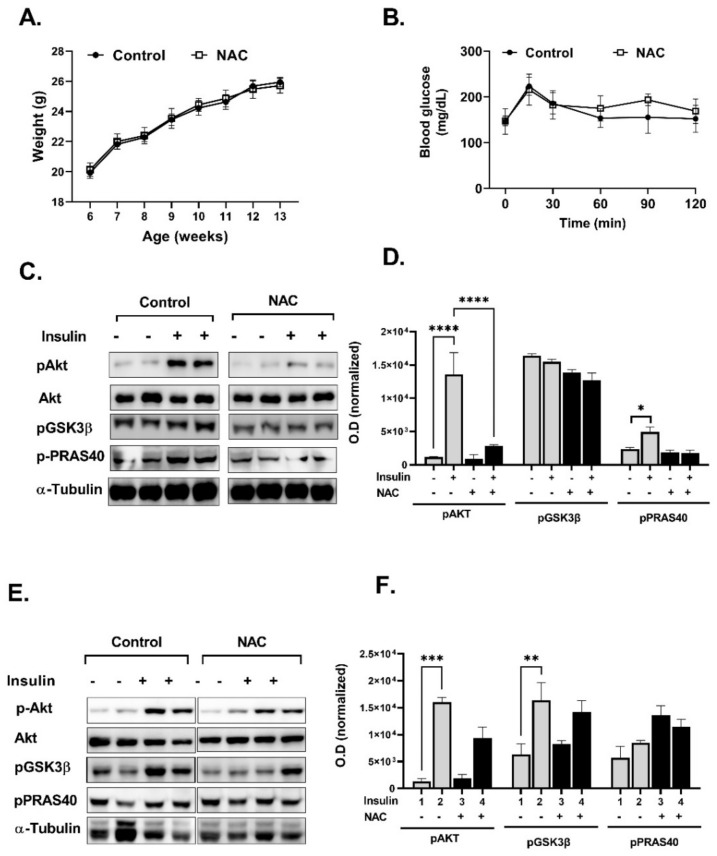
NAC attenuated insulin signaling in normoglycemic mice. C57BL/6 mice (*n* = 8, 6 weeks old) were given NAC (1200 mg/kg/day) for 6 weeks, as described in Methods. (**A**) Body weight was measured every week. (**B**) Glucose tolerance test (GTT) was performed at age of 12 weeks as described in Methods. Western blot analysis and densitometry analysis of proteins involved in insulin signaling pathway was performed in muscle (**C**,**D**) and liver (**E**,**F**). Optical density was normalized to the non-phosphorylated form or to housekeeping protein. The results are presented as the mean ± SEM. * *p* < 0.05, ** *p* < 0.005, *** *p* < 0.0005 and **** *p* < 0.0001 by one-way Anova (**A**) followed by Bonferroni’s post hoc test.

**Figure 10 ijms-23-02965-f010:**
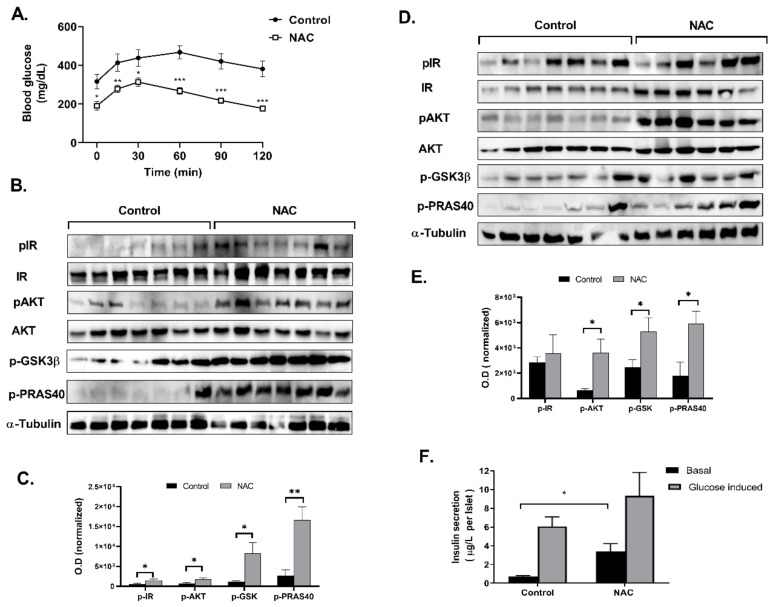
NAC improved glycemic control in diabetic KK-Ay mice. Mice (*n* = 10, 6 weeks old) were given NAC (1200 mg/kg/day) for 9 weeks, as described in Methods. (**A**) GTT was performed at the age of 15 weeks as described in Methods (*n* = 7). Western blot analysis and densitometry analysis of proteins involved in insulin signaling pathway was performed in muscle (**B**,**C**) and liver (**D**,**E**). Optical density was normalized to the non-phosphorylated form or to housekeeping protein. (**F**) Pancreatic islets were isolated, and GSIS was performed as described in Methods. The results are presented as the mean ± SEM. * *p* < 0.05, ** *p* < 0.005 and *** *p* < 0.0005 by one-way Anova followed by Bonferroni’s post hoc test.

## Data Availability

Not applicable.

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
