# Peer review of "Complexity of NAC Action as an Antidiabetic Agent: Opposing Effects of Oxidative and Reductive Stress on Insulin Secretion and Insulin Signaling"

_ijms, 2022, doi:10.3390/ijms23062965_

Round 1
Reviewer 1 Report
The work by Argaev-Frenkel and Rosenzweig aimed to address the specific effects of an antioxidant, N-acetylcysteine (NAC), in several mouse tissues. The study was motivated by their previous report demonstrating improvements in glucose tolerance and insulin sensitivity in diabetic mouse models receiving dietary supplementation of NAC. In this study, the authors employed different biological preparations including primary mouse islets and cultured cell lines (L6 myoblasts, 3T3 pre-adipocytes) to examine their responses to either oxidizing (H2O2) or reducing challenges (NAC). It was concluded that the effects of these agents can be complex, and vary with cell types or the mouse strains. While the data appeared to be solid, it is not obvious what insights have been gained from these in vitro studies to advance our understanding on the metabolic improvement of NAC in vivo; and where one might head for next from here. Another major concern relates to the supraphysiological concentrations of agents (high micromolar to millimolar) applied in the study. For instance, cells were treated with H2O2 or NAC in vitro at high micromolar to millimolar. It is unclear how these dosages relates to the in vivo situation where NAC was provided in the drinking water (1200 mg/Kg/day, through gavage?)
Author Response
Reviewer 1:
Comment: While the data appeared to be solid, it is not obvious what insights have been gained from these in vitro studies to advance our understanding on the metabolic improvement of NAC in vivo; and where one might head for next from here.
Response:
Thank you for the comment.
In this study, we demonstrate the tissue-specific effects of NAC, and showed that while pancreatic islets are highly sensitive to oxidative stress, and thus might benefit from NAC administration, insulin signaling and glucose transport are altered in response to NAC in myotubes and adipocytes. Thus, while it is expected to neutralize oxidative stress and be beneficial for insulin secretion, certain doses of NAC might ameliorate insulin response.
Supporting that, in-vivo experiments revealed beneficial outcome of NAC action when given to diabetic mice, although NAC given to non-diabetic mice presents a reduction in the transmission of insulin signaling. This data supports the importance of adjusting the AOX therapy to specific conditions, adjusting the dose to the specific redox alteration and avoiding unnecessary supplementation which might be harmful.
The clinical implications are the following:
- These data explain at least part of the failure of clinical studies with AOXs. Understanding the complexity of NAC action (and presumably other AOXs) is the first and essential step in order to develop strategies that overcome this complexity for gaining the benefits of NAC.
- Today, there are no commonly-accepted biomarkers and guidelines to the protocol of AOX supplementation. We assume that some clinical trial failed because of inappropriate dosage, either lower or higher than optimal. In the review of Schultz et al. (Oxidative stress and the use of antioxidants in diabetes: Linking basic science to clinical practice. Cardiovasc Diabetol 2005), it was revealed that: “Unfortunately, none of the studies to date effectively assessed the baseline oxidative stress of the enrolled patients using any of the commonly accepted markers of inflammation”. Our data suggest that a clear biomarker for the presence and severity of oxidative stress should be measured to identify the individuals that might gain advantage from AOX supplementation. In addition, we assume that redox state should be monitored at certain time-points during treatment in order to prevent overdose which might lead to a reductive stress and was shown in our research to ameliorate insulin signaling. Thus, future research should focus on the characterization of biomarkers for redox balance, for better personalized adjustment of the AOX treatment.
Accordingly, a sentence was added (lines 343-347), and the last two paragraphs of the MS were reorganized for a better clarification of this message (lines 458-477):
“In summary, by presenting the pros and cons of NAC action, this research explains at least part of the failure of clinical studies with AOXs. Understanding the complexity of NAC action is the first and essential step in order to develop strategies to overcome this obstacle for gaining the benefits of NAC. Such kind of double-edged sword of NAC action might be implicated in some other AOXs [82], but this should be further validated in future studies.
We conclude that NAC might either improve or impair glucose balance, depending on the specific dose, which, rather than being a fixed one, might depend on the specific redox status of the tissue, and the overall redox balance of the individual. With that understanding, it is clear that the lack of a measurable biomarker for the presence and severity of oxidative stress is a major obstacle in the implementation of NAC into clinical use. Today, there are no commonly-accepted biomarkers and guidelines to optimize a protocol of treatment with NAC or other AOXs for the treatment of T2D. Moreover, most clinical trials with AOXs lack assessment of baseline oxidative stress of participants [25]. Accordingly, future research should focus on the characterization of biomarkers for redox balance, to enable better personalized adjustment of the AOX treatment. Such biomarkers should be measured to identify the individuals that might gain advantage from AOX supplementation, and should be monitored at certain time-points during treatment in order to prevent reductive stress, which was shown in our research to ameliorate insulin signaling.”
Comment: Another major concern relates to the supraphysiological concentrations of agents (high micromolar to millimolar) applied in the study. For instance, cells were treated with H2O2 or NAC in vitro at high micromolar to millimolar. It is unclear how these dosages relates to the in vivo situation where NAC was provided in the drinking water (1200 mg/Kg/day, through gavage?)
Response:
Dose selection for in-vitro experiments was based on previous publications in the field. A large number of publications described the biological functions of NAC in various in-vitro systems. NAC (2.5-20 mM) downregulates VEGF production in melanoma cells (Redono P et al., Cytokine 2000). NAC (2-10 mM) induced differentiation and inhibited proliferation of Caco-2 and OVCAR3 (Parasassi T. et al., Cell Death Differ 2005), while inhibited myogenic differentiation (1-3 mM) (Rajasekaran NS. et al. Redox Biol 2020). Activation of Erk by NAC (8-10 mM) with the elevation of intracellular GSH was demonstrated in rat cardiac fibroblasts (Wang D., et al. J Biol Chem 1998). Suppression of NFkappaB was observed in pancreatic acinar cells treated by NAC (1-10 mM) (Kim H. et al., Free Radic. Biol. Med 2000) and in HeLa cells (10-30 mM) (Oka S. et al., FEBS Lett. 2000). Neutralization of oxidative stress in pancreatic islets was also performed in the dose range of millimolar (Takahashi H., et al. J Biol Chem 2004 and Roma LP., et al. Redox Report 2011). A large number of other publications, demonstrating several other biological functions of NAC are covered in the review of Samuni Y. et al. (The chemistry and biological activities of N-acetylcysteine. Biochim Biophysica Acta 2013).
In this study, we characterize the dose-dependent effect of NAC on several cell functions. We agree with the reviewer that the concentration scale includes high doses of NAC (in the milli-molar range), however, as mentioned, these doses are comparable to other studies. The results of our study demonstrate that these doses actually induce a reductive stress, as was also demonstrated in other publications (see references below). In fact, the main take-home message of this research is the conclusion that while pancreatic islets are highly sensitive to oxidative stress, insulin signaling in responsive tissues is impaired by reductive stress.
Similarly, H2O2-induced activation of signaling cascade was observed in concentration around 1 mM. We add citation of ref #1 (followed) to the discussion.
- Higaki Y et al., Oxidative stress stimulates skeletal muscle glucose uptake through a phosphatidylinositol 3-kinase-dependent pathway. Am J Physiol Endocrinol Metab. 2008.
- Hayes GR et al., Role of insulin receptor phosphorylation in the insulinomimetic effects of hydrogen peroxide. PNAS 1987)
- Heffetz D et al., The insulinomimetic agents H2O2 and vanadate stimulate protein tyrosine phosphorylation in intact cells. J Biol Chem 1990
- Valerie P et al., Redox modulation of global phosphatase activity and protein phosphorylation in intact skeletal muscle. J Physiol 2009
It is difficult to compare doses of NAC and H2O2 given in in-vitro experiments to the in-vivo situation. This is true for many agents due to the pharmacokinetics, including efficiency of absorption in the gut, stability, urinary excretion and so on. However, when dealing with oxidants and antioxidants, it is even more complicated to directly implicate in-vitro concentrations in in-vivo situations, because of the redox regulatory mechanisms which might neutralize oxidants and antioxidants. Thus, the aim of this study was to illustrate the principle of specific dose and tissue-specific effects to alterations in redox balance. Although the exact concentration might differ between in-vitro and in-vivo systems, the principle demonstrated here should be implicated in both.
NAC was given in drinking water, as mentioned in line 501 and as reported before in our previous studies (Michlin et al. IJMS 2020 and Falach-Malik et al. AM J Transl Res 2016). Average consumption of water was measured and NAC was dissolved in the drinking water so that the mice would receive NAC at the desired dose.
Reviewer 2 Report
This is a potentially interesting study of possible significance for diabetes. There are, however, several serious concerns.
- It is not at all clear in the Abstract what the connection between ROS and NAc is.
- The design of the paper is flawed: NAc is an antioxidant suggested to scavenge ROS. Yet, surprisingly, authors do not address whether e.g. NAc can protect islets against H2O2 functional suppression. These combined treatments must be carried out.
- These papers are pertinent and will need to be cited and discussed:
1: Roma LP, Oliveira CA, Carneiro EM, Albuquerque GG, Boschero AC, Souza KL.
N-acetylcysteine protects pancreatic islet against glucocorticoid toxicity.
Redox Rep. 2011;16(4):173-80. doi: 10.1179/1351000211Y.0000000006. PMID:
21888768; PMCID: PMC6837713.
2: Cappelli APG, Zoppi CC, Silveira LR, Batista TM, Paula FM, da Silva PMR,
Rafacho A, Barbosa-Sampaio HC, Boschero AC, Carneiro EM. Reduced glucose-induced
insulin secretion in low-protein-fed rats is associated with altered pancreatic
islets redox status. J Cell Physiol. 2018 Jan;233(1):486-496. doi:
10.1002/jcp.25908. Epub 2017 May 3. PMID: 28370189.
3: Takahashi H, Tran PO, LeRoy E, Harmon JS, Tanaka Y, Robertson RP.
D-Glyceraldehyde causes production of intracellular peroxide in pancreatic
islets, oxidative stress, and defective beta cell function via non-mitochondrial
pathways. J Biol Chem. 2004 Sep 3;279(36):37316-23. doi: 10.1074/jbc.M403070200.
Epub 2004 Jun 22. PMID: 15213233.
4: Oprescu AI, Bikopoulos G, Naassan A, Allister EM, Tang C, Park E, Uchino H,
Lewis GF, Fantus IG, Rozakis-Adcock M, Wheeler MB, Giacca A. Free fatty acid-
induced reduction in glucose-stimulated insulin secretion: evidence for a role
of oxidative stress in vitro and in vivo. Diabetes. 2007 Dec;56(12):2927-37.
doi: 10.2337/db07-0075. Epub 2007 Aug 23. PMID: 17717282.
5: Fu J, Zhang Q, Woods CG, Zheng H, Yang B, Qu W, Andersen ME, Pi J. Divergent
effects of sulforaphane on basal and glucose-stimulated insulin secretion in
β-cells: role of reactive oxygen species and induction of endogenous
antioxidants. Pharm Res. 2013 Sep;30(9):2248-59. doi: 10.1007/s11095-013-1013-8.
Epub 2013 Mar 7. PMID: 23468051; PMCID: PMC3718872.
6: Khaldi MZ, Elouil H, Guiot Y, Henquin JC, Jonas JC. Antioxidants N-acetyl-L-
cysteine and manganese(III)tetrakis (4-benzoic acid)porphyrin do not prevent
beta-cell dysfunction in rat islets cultured in high glucose for 1 wk. Am J
Physiol Endocrinol Metab. 2006 Jul;291(1):E137-46. doi:
10.1152/ajpendo.00145.2005. Epub 2006 Feb 7. PMID: 16464909.
7: Takatori A, Ishii Y, Itagaki S, Kyuwa S, Yoshikawa Y. Amelioration of the
beta-cell dysfunction in diabetic APA hamsters by antioxidants and AGE inhibitor
treatments. Diabetes Metab Res Rev. 2004 May-Jun;20(3):211-8. doi:
10.1002/dmrr.428. PMID: 15133752.
8: Ammon HP, Hehl KH, Enz G, Setiadi-Ranti A, Verspohl EJ. Cysteine analogues
potentiate glucose-induced insulin release in vitro. Diabetes. 1986
Dec;35(12):1390-6. doi: 10.2337/diab.35.12.1390. PMID: 3533685.
9: Tanaka Y, Gleason CE, Tran PO, Harmon JS, Robertson RP. Prevention of glucose
toxicity in HIT-T15 cells and Zucker diabetic fatty rats by antioxidants. Proc
Natl Acad Sci U S A. 1999 Sep 14;96(19):10857-62. doi: 10.1073/pnas.96.19.10857.
PMID: 10485916; PMCID: PMC17973.
- Statistics: Were data normally distributed? Were any power calculations made? The number of observations (n) is nowhere stated. This is unacceptable.
Minor
- 96: Explain ”naive” and ”ICR”.
Author Response
Reviewer 2:
Comment:
It is not at all clear in the Abstract what the connection between ROS and NAc is.
Response:
Abstract was reorganized and corrected accordingly.
Comment:
The design of the paper is flawed: NAc is an antioxidant suggested to scavenge ROS. Yet, surprisingly, authors do not address whether e.g. NAc can protect islets against H2O2 functional suppression. These combined treatments must be carried out.
Response:
The properties of NAC as an AOX that efficiently neutralizes ROS is well characterized in previous publications. It was already shown that NAC counteracts the damage of ROS, generated in response to H2O2 (Khaldi et al., Endocrinol Metab 2006), dexamethasone (Roma et al., Redox Rep 2011) or chronic glucotoxicity (Takahashi et al., J Biol Chem 2004). These publications are cited (according to the reviewer’s recommendation) in lines 440-443.
Accordingly, this kind of experiment was not performed again by us. However, differing from previous publications demonstrating the potency of NAC as an AOX counteracting the negative consequence of oxidizing conditions, in this study we tried to find out whether NAC might exert either positive or negative outcomes on its own, comparing the response of the cells to increased oxidizing and reductive conditions. Accordingly, increasing concentrations of NAC and H2O2 were used in either pancreatic islets or adipocytes and myotubes, showing that high NAC doses induced reductive stress in view of insulin signaling, but not in view of insulin secretion, while H2O2, even at low doses, impaired insulin secretion while promoting insulin signaling.
Comment:
These papers are pertinent and will need to be cited and discussed:
1: Roma LP, Oliveira CA, Carneiro EM, Albuquerque GG, Boschero AC, Souza KL.
N-acetylcysteine protects pancreatic islet against glucocorticoid toxicity.
Redox Rep. 2011;16(4):173-80. doi: 10.1179/1351000211Y.0000000006. PMID:
21888768; PMCID: PMC6837713.
2: Cappelli APG, Zoppi CC, Silveira LR, Batista TM, Paula FM, da Silva PMR,
Rafacho A, Barbosa-Sampaio HC, Boschero AC, Carneiro EM. Reduced glucose-induced
insulin secretion in low-protein-fed rats is associated with altered pancreatic
islets redox status. J Cell Physiol. 2018 Jan;233(1):486-496. doi:
10.1002/jcp.25908. Epub 2017 May 3. PMID: 28370189.
3: Takahashi H, Tran PO, LeRoy E, Harmon JS, Tanaka Y, Robertson RP.
D-Glyceraldehyde causes production of intracellular peroxide in pancreatic
islets, oxidative stress, and defective beta cell function via non-mitochondrial
pathways. J Biol Chem. 2004 Sep 3;279(36):37316-23. doi: 10.1074/jbc.M403070200.
Epub 2004 Jun 22. PMID: 15213233.
4: Oprescu AI, Bikopoulos G, Naassan A, Allister EM, Tang C, Park E, Uchino H,
Lewis GF, Fantus IG, Rozakis-Adcock M, Wheeler MB, Giacca A. Free fatty acid-
induced reduction in glucose-stimulated insulin secretion: evidence for a role
of oxidative stress in vitro and in vivo. Diabetes. 2007 Dec;56(12):2927-37.
doi: 10.2337/db07-0075. Epub 2007 Aug 23. PMID: 17717282.
5: Fu J, Zhang Q, Woods CG, Zheng H, Yang B, Qu W, Andersen ME, Pi J. Divergent
effects of sulforaphane on basal and glucose-stimulated insulin secretion in
β-cells: role of reactive oxygen species and induction of endogenous
antioxidants. Pharm Res. 2013 Sep;30(9):2248-59. doi: 10.1007/s11095-013-1013-8.
Epub 2013 Mar 7. PMID: 23468051; PMCID: PMC3718872.
6: Khaldi MZ, Elouil H, Guiot Y, Henquin JC, Jonas JC. Antioxidants N-acetyl-L-
cysteine and manganese(III)tetrakis (4-benzoic acid)porphyrin do not prevent
beta-cell dysfunction in rat islets cultured in high glucose for 1 wk. Am J
Physiol Endocrinol Metab. 2006 Jul;291(1):E137-46. doi:
10.1152/ajpendo.00145.2005. Epub 2006 Feb 7. PMID: 16464909.
7: Takatori A, Ishii Y, Itagaki S, Kyuwa S, Yoshikawa Y. Amelioration of the
beta-cell dysfunction in diabetic APA hamsters by antioxidants and AGE inhibitor
treatments. Diabetes Metab Res Rev. 2004 May-Jun;20(3):211-8. doi:
10.1002/dmrr.428. PMID: 15133752.
8: Ammon HP, Hehl KH, Enz G, Setiadi-Ranti A, Verspohl EJ. Cysteine analogues
potentiate glucose-induced insulin release in vitro. Diabetes. 1986
Dec;35(12):1390-6. doi: 10.2337/diab.35.12.1390. PMID: 3533685.
9: Tanaka Y, Gleason CE, Tran PO, Harmon JS, Robertson RP. Prevention of glucose
toxicity in HIT-T15 cells and Zucker diabetic fatty rats by antioxidants. Proc
Natl Acad Sci U S A. 1999 Sep 14;96(19):10857-62. doi: 10.1073/pnas.96.19.10857.
PMID: 10485916; PMCID: PMC17973.
Response:
We appreciate the comment. These publications were discussed and cited in the revised MS (Lines 354-372, lines 440-443).
Comment:
Statistics: Were data normally distributed? Were any power calculations made? The number of observations (n) is nowhere stated. This is unacceptable.
Response:
Thank you for the comment. Data were normally distributed. Power analysis was performed. Power=0.8, alpha=0.05.
Number of observations was added to the figure legends.
Comment:
96: Explain ”naive” and ”ICR”.
Response:
The term “naïve” was replaced with “untreated”.
ICR mice: This is the most widely-used outbred mouse, generated from the original two male and seven female swiss albino non-inbred mice. It is widely used in biomedical research (Kim et al., Annual tendency of research papers used ICR mice as experimental animals in biomedical research fields. Lab Anim Res 2017). An explanation was added to text (line 515).
Round 2
Reviewer 1 Report
The authors have partially addressed my concerns. While the dosage of NAC and H2O2 used in this study is in the range used in other studies, the high concentration nonetheless still raises the concern of the physiological relevance. This caveat should be mentioned in the discussion. Finally, NAC was provided in the drinking water. In the Method section, the authors need to provide the typical NAC concentration in the drinking water; and the average volume of water consumption, including both NAC water and non-treated water (control).
Reviewer 2 Report
Revision acceptable.
Author Response
Reviewer had no comments.